# Complete Chloroplast Genome and Phylogenomic Analysis of *Davallia trichomanoides* (Polypodiaceae)

**DOI:** 10.3390/genes16111310

**Published:** 2025-11-01

**Authors:** Yingying Wang, Ziqi Xiang, Keqin Liu, Yuan Lin, Siyuan Dong

**Affiliations:** 1College of Liangshan, Lishui University, Lishui 323000, China; wangyy@lsu.edu.cn (Y.W.); xzq010507@lsu.edu.cn (Z.X.); lkq@lsu.edu.cn (K.L.); 2Institute of Synthetic Biology, Shenzhen Institute of Advanced Technology, Chinese Academy of Sciences, Shenzhen 518055, China; 3China National Research Institute of Food and Fermentation Industries Co., Ltd., Beijing 100015, China

**Keywords:** Polypodiaceae, *Davallia trichomanoides*, chloroplast genome, comparative plastome, genetic diversity, phylogenetic analysis

## Abstract

**Background/Objectives**: Chloroplast genomes (plastomes) are valuable for fern systematics, yet the epiphytic lineages have remained underexplored. **Methods**: The *Davallia trichomanoides* plastome was de novo assembled from Illumina data and annotated. **Results**: The plastome measures 154,217 bp with a GC content of 40.82% and contains 115 genes. Comparative analysis reveals two inverted repeat (IR) size classes (~24.0–24.6 kb vs. ~27.4–27.5 kb) and lineage-specific shifts at the IR junctions. For instance, the *ndhF* gene remains in the small single copy (SSC) region in *D. trichomanoides* and *Drynaria acuminata*, but it crosses into the IRb region in other species. We observed nucleotide diversity hotspots in the large single copy (LSC) and SSC regions. The IR regions are highly conserved. The ratios of nonsynonymous to synonymous substitutions (Ka/Ks) are mostly less than 1, indicating purifying selection. Phylogenetic analysis places *D. trichomanoides* as the sister to *D. acuminata*. **Conclusions**: This study highlights the stable plastome structure of *D. trichomanoides* and identifies candidate loci for barcoding. It also supports the stable placement of *Davallia* within the epiphytic Polypodiineae.

## 1. Introduction

In land plants, chloroplast genomes (plastomes) are usually circular and divided into a large single-copy (LSC), a small single-copy (SSC), and two inverted repeats (IRs). This stable structure underlies the broad application of plastid DNA in systematics and comparative genomics, as well as in various applied contexts ranging from phylogenetic studies [1] to population-level analyses [2]. Although structurally conservative, most changes in length or gene order reflect small IR expansions or contractions, or local inversions, and the IRs are usually the most conserved parts [3]. This IR-linked stability and lower substitution rate are well documented across land-plant plastomes [4].

Ferns provide a complementary macroevolutionary view to seed plants. Broad surveys show that fern plastomes keep a conservative gene set but display lineage-specific structural changes—notably large inversions, shifts at IR boundaries, and occasional gene losses. These features help resolve deep nodes and assess rate heterogeneity in fern radiations [5]. Recent syntheses underscore that such structural variation is often clade-specific [6]. Within Polypodiales, the most species-rich fern order, plastid phylogenomics has clarified relationships in Polypodiaceae and revealed distinct dynamics in epiphytic lineages [7]. Comparative work in Polypodiaceae shows overall collinearity alongside local structural shifts [8].

At finer scales, junction architecture and repeat landscapes carry much of the among-lineage signal. Specifically, border-proximal genes (e.g., *ycf1*, *ndhF*, *rps19*/*rpl2*) are repeatedly linked to IR expansion or contraction. These structural changes directly alter plastome size and collinearity. Non-coding spacers in the LSC/SSC usually diverge more than coding regions or the IRs, yielding candidate barcodes and phylogenetically informative windows [9,10]. Repeats add another axis of variation: dispersed repeats (forward, reverse, complement, palindromic) and mobile open reading frames (ORFs) can interrupt otherwise conserved synteny, whereas simple sequence repeats (SSRs) provide high-copy, population-level markers [4]. These patterns also appear in recent fern and angiosperm plastome comparisons [6,8].

Recent complete plastomes for *Drynaria acuminata* [11], *Lemmaphyllum intermedium* [12], *Leptochilus*/*Lepisorus* spp. [13], and *Neocheiropteris* [14] outline two IR size classes and local SSC/IR boundary shifts. Nevertheless, taxonomic coverage remains patchy. Key questions also persist regarding the extent and pace of junction movement, repeat-mediated micro-rearrangements, and the distribution of divergence hotspots. To our knowledge, a complete plastome for *D. trichomanoides* has not yet been reported.

*D*. *trichomanoides* is a widespread Asian epiphyte in the *Davalliaceae* complex and is often analyzed alongside Polypodiaceae in plastid phylogenomic matrices. A high-quality plastome for *D. trichomanoides* fills a taxon gap and enables side-by-side comparisons of IR junctions and repeat profiles across epiphytic clades. It also tests whether features reported in *Drynaria*, *Lemmaphyllum*, *Lepisorus*/*Leptochilus*, and *Neocheiropteris* extend to *Davallia* or reflect lineage-specific histories [7,11,12,13,14]. Adding *D. trichomanoides* lets us assign its IR size class and examine SSC/IR boundary behavior within a consistent comparative framework.

In this study, we present the complete plastome of *D. trichomanoides* and compare it with related epiphytic ferns. We report genome size and GC content and assess collinearity. We map the four junctions (JLB, JSB, JSA, JLA) to evaluate IR expansion/contraction and border-gene shifts. We catalog dispersed repeats and simple SSRs. We profile nucleotide diversity (Pi values) to flag hypervariable windows. We infer a plastome phylogeny and compare it with recent Polypodiaceae reconstructions. Placing *D. trichomanoides* in this literature-based framework lets us test whether Polypodiaceae patterns of junction dynamics and repeat signatures extend to *Davalliaceae* s.l., and it highlights candidate loci for barcoding and evolutionary analyses.

## 2. Materials and Methods

### 2.1. Sampling and Sequencing

We collected fresh leaves of *D*. *trichomanoides* near Huaihua, Hunan, China (26°14′54″ N, 109°51′40″ E). We deposited a voucher (LSU_20250611) in the Herbarium of Lishui University. We extracted total DNA from leaves using the Rapid Plant Genomic DNA Isolation Kit (Sangon Biotech, Shanghai, China). The extract contained both nuclear and plastid DNA; in downstream analyses, we specifically targeted the plastome for assembly. We prepared 150-bp paired-end (PE) libraries and sequenced them on an Illumina HiSeq 2500 (San Diego, CA, USA), yielding 93,481,172 reads (13.93 GB) for subsequent assembly.

### 2.2. Assembly and Annotation of the Plastome

We generated 13.93 GB of Illumina 150-bp PE reads for plastome assembly. We trimmed adapters and low-quality bases with Trimmomatic v0.39 [15]. We assembled the plastome with GetOrganelle v1.5 [16] and retained the circular contig as the final plastome. We annotated genes with GeSeq v2.03 [17] and CPGAVAS2 v2.1.0 [18], identifying protein-coding genes (PCGs), rRNAs, tRNAs, IR boundaries, and genome orientation. We then checked and manually refined annotations in CPGView (http://www.1kmpg.cn/cpgview; accessed on 28 October 2025) [19], confirming exon–intron structure and cis-/trans-spliced genes. The finalized plastome of *D. trichomanoides* is deposited in GenBank (PX170254).

### 2.3. Plastome Characteristics

We calculated overall GC content from the finalized plastome in EditSeq v7.1.0 [20]. We generated the circular chloroplast map with OGDRAW v1.3.1 [21] using the curated annotation, showing the LSC, SSC, and IR partitions with standard gene-feature tracks.

### 2.4. SSR Analyses

We screened interspersed repeats with REPuter (https://bibiserv.cebitec.uni-bielefeld.de/reputer/, accessed on 28 October 2025) [22], searching for forward, reverse, complement, and palindromic matches ≥ 30 bp and ≤ 100 bp, and allowing up to three mismatches (Hamming distance ≤ 3). We then identified SSRs using MISA v2.1 [23] with motif thresholds of 10, 5, 4, 3, 3, and 3 for mono-, di-, tri-, tetra-, penta-, and hexanucleotides, respectively. We treated loci with interruptions no longer than 100 bp as compound SSRs.

### 2.5. Analysis of Codon Preference

We calculated codon-usage statistics and relative synonymous codon usage (RSCU) in CodonW v1.4.4 [24] using all annotated PCGs, the plant plastid genetic code (translation table 11), and excluding stop codons. We then visualized the RSCU profiles for comparative interpretation.

### 2.6. Analysis of IR Boundary

We used IRscope v3.1 [25] to visualize the four plastome junctions (JLB, JSB, JSA, JLA) in *D. trichomanoides* and four related taxa: *D*. *acuminata*, *Neocheiropteris ovata*, *L*. *intermedium*, and *Lepisorus microphyllus*. We chose *D*. *acuminata*, *N*. *ovata*, *L*. *intermedium*, and a representative of *Lepisorus* because these epiphytic Polypodiineae lineages are the closest and best-sampled plastid comparators relevant to *Davallia*’s placement; complete, publicly available plastomes allow for like-for-like analyses; and together, they encompass the two predominant IR size classes and the characteristic SSC/IR junction shifts documented for allied Polypodiaceae. We loaded curated GenBank annotations into IRscope, generated junction diagrams, and recorded junction coordinates, gene overlaps, and partial-gene incursions across the LSC, SSC, IRa, and IRb regions.

### 2.7. Analysis of Selection Pressure

We used the *D. trichomanoides* plastome as the reference and four related taxa as queries. We extracted CDS and proteins in PhyloSuite v1.2.1 [26]. We identified putative homologs with BLASTP (BLAST+ v2.10.1) against the *D. trichomanoides* proteome and kept the top hit per gene. We aligned proteins with MAFFT v7.526 [27] and back-translated to codon-aware nucleotide alignments (PAL2NAL-style). Before rate estimation, we removed problematic ORFs (frameshifts, ambiguous bases, and internal stops) and masked low-confidence columns with Gblocks (default). We estimated pairwise nonsynonymous (Ka) and synonymous (Ks) with KaKs_Calculator v3.0 [28] and summarized Ka/Ks (ω) per gene. To reduce artifacts, we reported ω as NA when Ks = 0, did not interpret genes with saturated divergence (Ks > 2), and did not claim positive selection from ω > 1 without additional site-model support.

### 2.8. Comparative Analysis of Plastomes

We aligned four related plastomes to the *D. trichomanoides* reference in mVISTA (Shuffle-LAGAN) (http://genome.lbl.gov/vista/mvista/submit.shtml, accessed on 28 October 2025) [29]. We then computed Pi values in DnaSP v6 [30] using a 600-bp window and 200-bp step (gaps/missing treated as missing). We flagged windows above the 95th percentile of the Pi value as hypervariable for downstream reporting.

### 2.9. Phylogenetic Relationship Analysis

We analyzed complete plastomes from 18 fern taxa and set *Lepidomicrosorium hederaceum* as the outgroup. We aligned sequences with MUSCLE v3.8.31 [31]. We inferred a maximum-likelihood tree in RAxML v8.2.12 [32] under GTR+G+I with 1000 rapid bootstrap replicates and a best-scoring maximum-likelihood search. We formatted and annotated the topology and node supports in R using ggtree v3.2.1 [33].

### 2.10. Comparative Synteny Analysis

We aligned the circularized plastomes with MUMmer v4 [34] after re-orienting each genome at JLB (LSC–IRb). To avoid duplicate hits, we retained only one IR copy per genome. We filtered alignments to identity ≥ 80% and length ≥ 500 bp, and used strand to classify forward vs. reverse matches. We visualized the filtered blocks in Circos v0.69 [35], highlighting the LSC/SSC/IR partitions, and used these plots to assess collinearity and detect short, localized inversions.

## 3. Results

### 3.1. Plastome Assembly and Genome Features

Illumina PE sequencing generated 93,481,172 reads (13.93 GB) with high base accuracy (Q20 = 98.84%, Q30 = 95.94%; Appendix A). The plastome of *D*. *trichomanoides* was de novo assembled and circularized with GetOrganelle. Mapping all clean reads back to the assembly showed uniform coverage (mean depth 535.65 ×) with 100% (Appendix A). The plastome of *D*. *trichomanoides* is circular and quadripartite (Figure 1), comprising a LSC region of 84,713 bp, an SSC region of 21,436 bp, and two inverted repeats (IRa/IRb) of 24,034 bp each. The overall GC content is 40.82% (IRs 44.38%, LSC 39.86%, SSC 36.62%).

The plastome annotation recovered 115 genes in total, comprising 75 PCGs, 32 tRNA genes, and 8 rRNA genes (Table 1). Sixteen genes were intron-bearing in this genome: *trnG-UCC*, *atpF*, *rpoC1*, *ycf3*, *trnL-UAA*, *trnV-UAC*, *rps12*, *clpP*, *rpl2*, *trnT-UGU*, *trnA-UGC*, *trnI-GAU*, *ndhA*, and the IR-duplicated copies *trnT-UGU-2*, *trnA-UGC-2*, *trnI-GAU-2* (Table 2). Among these, 10 genes contained a single intron (*trnG-UCC*, *atpF*, *rpoC1*, *trnL-UAA*, *trnV-UAC*, *rpl2*, *trnT-UGU*, *trnA-UGC*, *trnI-GAU*, and *ndhA*), whereas *clpP*, *rps12*, and *ycf3* each harbored two introns. Consistent with typical fern plastomes, *rps12* was trans-spliced. The remaining intron-bearing loci represented cis-splicing genes. These cis-splicing genes included *atpF*, *rpoC1*, *clpP*, *ycf3*, *rpl2*, *ndhA*, and six intron-containing tRNA genes (*trnG-UCC*, *trnL-UAA*, *trnV-UAC*, *trnT-UGU*, *trnA-UGC*, and *trnI-GAU*) (Table 2).

### 3.2. Repeat Sequences and SSR Analysis

Repeats detected with REPuter showed a strongly right-skewed length distribution: most fell in the 20–29 bp bin, counts declined sharply with increasing length, and only sporadic cases were ≥ 50 bp. By type, palindromic and forward repeats dominated, reverse repeats were secondary, and complementary repeats were rare or absent (Figure 2A). The SSR set comprised mono-, di-, tri-, and tetranucleotide motifs; penta- and hexanucleotide SSRs were not detected. Mononucleotide SSRs were most abundant, followed by dinucleotides; trinucleotides and tetranucleotides were less common. Among mononucleotides, A/T motifs predominated; among dinucleotides, AT/TA prevailed; trinucleotides were enriched for AAT/ATT and AAG/CTT types (Figure 2B).

### 3.3. Codon Usage of Plastome

RSCU profiling of plastid PCGs in *D. trichomanoides* revealed a third-position bias toward A/U. In two- and four-fold degenerate families, A/U-ending codons were generally over-represented (RSCU > 1), whereas most G/C-ending counterparts were under-represented (RSCU < 1). The same pattern held for the six-fold families (Leu, Ser, Arg), in which A/U-ending codons showed higher RSCU than G/C-ending codons. Single-codon families—Met (AUG) and Trp (UGG)—had RSCU = 1.0 (Figure 3).

### 3.4. IR Boundary Comparison

Comparative mapping of the four junctions (JLB, JSB, JSA, and JLA) across five Polypodiaceae plastomes shows a generally conserved quadripartite organization with modest size variation (Figure 4). Total plastome sizes span 151,591–158,029 bp. The LSC ranges from 80,661–84,713 bp, the SSC from 21,436–21,797 bp, and each IR from 24,034–27,494 bp. Two IR size classes are apparent: ~24.0–24.6 kb in *D. trichomanoides* (24,034 bp), *D. acuminata* (24,621 bp) and *N. ovata* (24,609 bp), versus ~27.4–27.5 kb in *L. intermedium* (27,421 bp) and *L. microphyllus* (27,494 bp), indicating lineage-specific IR expansion in the latter pair.

At JLB (LSC–IRb), *trnI* lies consistently close to the border (~49–73 bp across taxa), indicating minimal LSC→IRb drift. At JSB (IRb–SSC), *ndhF* shows lineage-specific shifts: in *D. trichomanoides* and *D. acuminata* it remains in the SSC at similar distances (~1.8 kb) from JSB, whereas in *N. ovata*, *L. intermedium*, and *L. microphyllus*, the 3′ end of *ndhF* crosses the junction into IRb by ~43 bp, 18 bp, and 14 bp, respectively. In contrast, JSA (SSC–IRa) is highly conserved, with *chlN* situated ~900 bp from the junction in all species. At JLA (IRa–LSC), *matK* in *D. trichomanoides* is 1475 bp from the border; *D. acuminata* appears similar, whereas in *N. ovata*, *L. intermedium*, and *L. microphyllus*, *matK* lies closer to JLA. The duplicated *ndhB* copies about JLA (~1 bp) in the compared taxa.

### 3.5. Comparative Analysis of Plastomes

The mVISTA program (reference *D. trichomanoides*) shows high overall similarity among the five plastomes (Figure 5). Consistent with plastome organization, regions corresponding to the IRs appear more conserved than the LSC/SSC, and non-coding spacers vary more than coding sequences. Conserved peaks cluster around photosynthesis- and ribosome-related gene blocks (e.g., *psb*/*psa*/*atp*/*rpl*/*rps*), whereas most divergence is localized to intergenic spacers across the LSC and SSC.

Using DnaSP sliding windows, nucleotide diversity across five plastomes shows low overall variability in the IRs and elevated peaks in the LSC/SSC (Figure 6). Pi values span ~0–0.059, with the highest signal immediately upstream of *psal*-*ycf4* (58382–59402 bp; Pi = 0.059). Recurrent hypervariable segments (Pi ≥ 0.046) occur around *trnS-GCU*, *trnC-GCA*, and *trnG-GCC* block, the *psal*-*ycf4* region, and within the SSC near the IRb boundary (e.g., 104.6–106.2 kb, 108.7–110.0 kb, the latter at *trnP-GGG*). In contrast, IR-embedded loci such as *rrn23*/*rrn16* and *psbA* remain highly conserved (Pi ≈ 0–0.01).

### 3.6. Selection Pressure Analysis

Ks and Ka substitution rates were estimated for plastome PCGs. Gene-wise Ka/Ks ratios ranged from 0.00 to 0.88 (mean 0.15). No gene exhibited Ka/Ks ≥ 1.0, indicating uniformly low ratios across loci (Figure 7).

### 3.7. Comparative Synteny Analysis

The circos synteny plot indicates extensive forward collinearity between the focal plastome and closely related genomes, with paired signals across the IRs and only short, localized reverse matches in the single-copy regions. These patterns support a largely conserved plastome architecture with minor micro-rearrangements (Figure 8).

### 3.8. Phylogenetic Analysis

A maximum-likelihood tree of 18 fern plastomes (*L. hederaceum* as outgroup) recovered a backbone with uniformly high support (most nodes bootstrap ≥ 99; many at 100). *D. trichomanoides* formed a strongly supported lineage sister to *D. acuminata* (bootstrap = 100), clarifying its position relative to allied polypodiaceous genera. Monophyly was recovered for the sampled clades of *Pyrrosia* (multiple accessions clustering with bootstrap = 100) and *Lemmaphyllum* (*L. intermedium* + *L. carnosum*, bootstrap = 100), with *Neocheiropteris* accessions grouping together and distinct from *Microsorum*/*Lepisorus* lineages, all with maximal or near-maximal support. Overall, the topology is congruent with current taxonomy and indicates limited conflict among plastome histories, while highlighting the close affinity of *Davallia* and *Drynaria* within this sampling (Figure 9).

## 4. Discussion

### 4.1. Plastome Organization in D. trichomanoides

The *D. trichomanoides* plastome matches the canonical quadripartite layout and size typical of leptosporangiate ferns, with GC content and gene repertoire within the range for Polypodiales [36,37,38]. The IRs are far less variable than the LSC/SSC regions, and gene order is largely conserved, underscoring the stabilizing role of the duplicated IRs [36,37]. Overall, these features agree with broad syntheses showing that most land-plant plastomes—including ferns—are structurally conservative, with lineage-specific departures concentrated at single-copy/IR junctions and in non-coding spacers [36,37,38].

### 4.2. IR Boundary Dynamics

Across the five plastomes, we recovered two IR size classes and small, taxon-specific junction shifts. At JLB, *trnI* sits at the border in all taxa, indicating little LSC→IRb drift. JSB is heterogeneous: in *D. trichomanoides* and *D. acuminata*, *ndhF* remains in the SSC at a similar offset, whereas in *N. ovata*, *L. intermedium*, and *L. microphyllus*, the 3′ end extends a few tens of bp into IRb. Such minor, repeated incursions of *ndhF* or *ycf1* across JSB/JSA are a common sign of IR expansion/contraction [7,36]. These small junction shifts explain much of the length variation observed among otherwise collinear plastomes. At JLA, *matK* lies ~1.5 kb inside the LSC in *D. trichomanoides*, marking a species-specific terminal shift without broader disruption—again consistent with incremental boundary movement rather than large rearrangements [7,36].

### 4.3. Patterns of Sequence Divergence and Marker Potential

Sliding-window nucleotide diversity (Pi values) shows that divergence concentrates in LSC/SSC intergenic spacers and a few coding genes, whereas the IRs are conserved [36,37,38]. The top windows (e.g., *psaI–ycf4*, *trnS–trnG*, *trnC–trnG*, and SSC-proximal segments) are practical targets for marker development in *Davallia* and its relatives. Peaks near *ycf1* and *ndhF* match broader evidence that these loci are among the most variable plastid genes at low taxonomic levels and can outperform classical barcodes [39,40]. Including these hotspots in multi-locus panels should improve resolution for populations and species complexes in epiphytic ferns [39,40].

### 4.4. Repeats and SSRs

The repeat landscape is dominated by forward and palindromic repeats, as in many fern and seed-plant plastomes. The SSR set is A/T-rich, with mononucleotide runs most abundant—patterns widely reported for plastomes [36,41]. SSRs, especially A/T homopolymers in the LSC/SSC regions, are useful for population studies and for building genotyping panels, provided homopolymer length variation is validated across platforms [41]. This A/T bias in plastid SSRs recurs across diverse plant groups, underscoring its generality and value for intraspecific work [41].

### 4.5. Selection on PCGs

Ka/Ks ratios are mostly <1, indicating pervasive purifying selection on plastid PCGs, a pattern widely reported for land plants and ferns [38,41]. Occasional locus-specific elevations are more plausibly due to relaxed constraint, lineage-specific rate variation, or local alignment uncertainty than to broad positive selection; substantiating adaptive signals would require targeted site-model tests and denser taxon sampling [38,41].

### 4.6. Collinearity and Structural Stability

Whole-plastome collinearity is high, with only short, localized reverse matches in the single-copy regions and no large inversions like those reported in deeper fern lineages [37,42]. This supports a “conservative-with-micro-edits” model for *D. trichomanoides*: genome-wide synteny persists, while small changes arise via repeat-mediated events and incremental junction shifts [37,42].

### 4.7. Phylogenetic Placement and Taxonomic Congruence

The plastome maximum-likelihood tree places *D. trichomanoides* as sister to *D. acuminata* with strong support, matching recent plastid phylogenomic studies in Polypodiaceae that stabilize epiphytic lineages [7]. The small boundary differences noted above do not disrupt the plastome-scale signal at this depth. The plastome is a single, largely non-recombining and typically uniparentally inherited locus. Consequently, it records the maternal history and can be affected by processes like introgression or incomplete lineage sorting. We therefore interpret this topology as representing the plastid history of the group. Future work will add nuclear target-capture data to test plastid–nuclear concordance and refine relationships.

### 4.8. Ecological Context of Epiphytic Ferns

Epiphytic canopies impose rapid light swings and intermittent water supply, and many epiphytic ferns are poikilohydric with quick post-drying recovery [43]. In this context, several genomic features of *D. trichomanoides* are consistent with stabilizing selection on core photosynthetic machinery [8,44]. These features include its high plastome collinearity, strong IR conservation (evidenced by relatively low Pi values compared to LSC/SSC), and gene-wise Ka/Ks ratios consistently below one. The IR likely enhances plastome stability and increases rRNA operon dosage, supporting translation under stress [45]. Retention of chloroplast *ndh* genes is also ecologically plausible, as the NDH complex protects PSI and sustains cyclic electron flow under fluctuating light and water deficit [46].

## 5. Conclusions

This study characterizes the plastome of *D. trichomanoides* and, in comparison with related epiphytic ferns, shows lineage-specific differences at inverted-repeat/single-copy junctions that explain modest size variation while preserving overall collinearity. Small IR-border shifts—often involving *ycf1* and *ndhF*—fit the common expansion–contraction model that drives plastome length change. The plastome tree places *D. trichomanoides* with *D. acuminata* with strong support, consistent with recent Polypodiaceae plastid phylogenies and clarifying relationships among epiphytic lineages. Dispersed repeats and A/T-rich cpSSRs, together with Pi values peaks concentrated in LSC/SSC, provide practical loci for barcoding and population studies; *ycf1* and adjacent spacers remain high-value targets. Going forward, we will use these hotspots (e.g., *ycf1-*, *ndhF*-, and *psaI*–*ycf4* regions) to build multi-locus barcodes and test whether SSR variation tracks geography, informing germplasm characterization and conservation in *Davallia*. These steps extend the present plastome framework from systematics to mechanism-focused and population-scale questions in epiphytic Polypodiineae.

## Figures and Tables

**Figure 1 genes-16-01310-f001:**
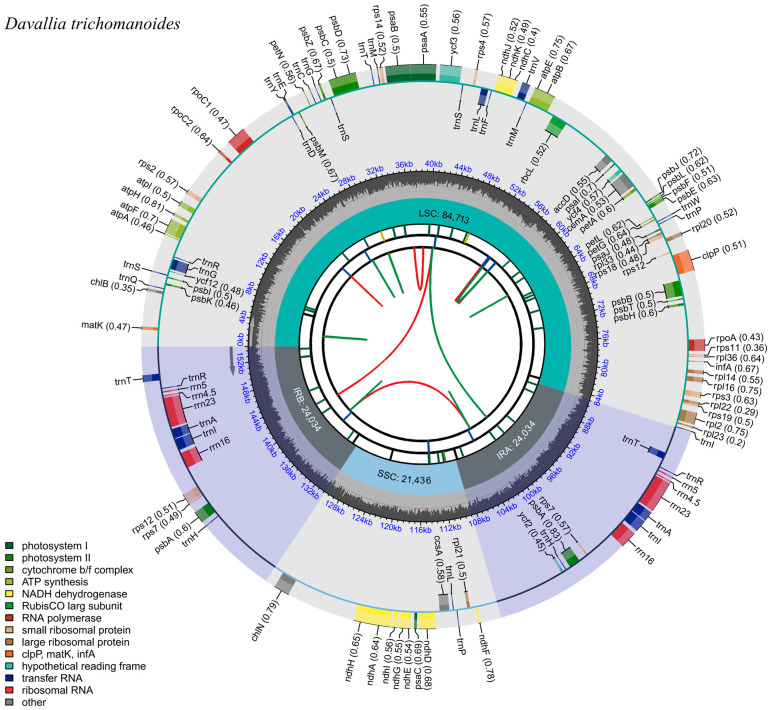
Circular plastome map of *Davallia trichomanoides* (154,217 bp). The plastome is quadripartite, comprising the large single-copy (LSC), small single-copy (SSC), and two inverted repeats (IRa/IRb; lengths indicated). Genes are colored by functional category; the inner ring shows GC content.

**Figure 2 genes-16-01310-f002:**
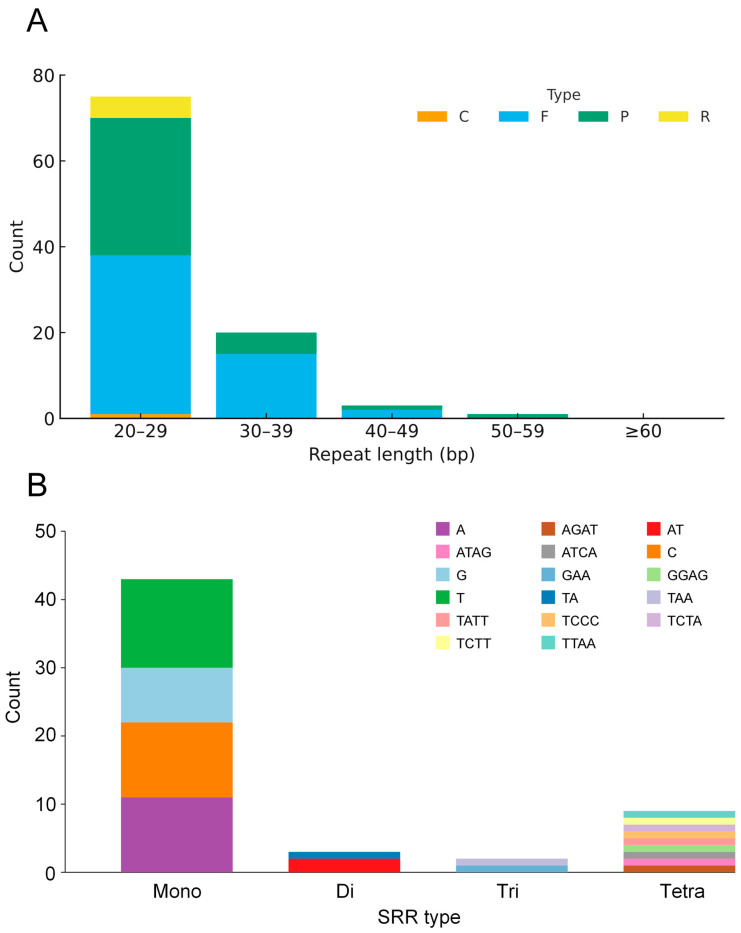
Analysis of repetitive elements in the *Davallia trichomanoides* plastome. (**A**) Counts of four types of interspersed repeats (forward, reverse, complement, and palindromic), identified with a size threshold of ≥30 bp and allowing ≤3 mismatches. (**B**) Counts of simple sequence repeats (SSRs) categorized by the length of their repeating motif (mono- to hexanucleotide).

**Figure 3 genes-16-01310-f003:**
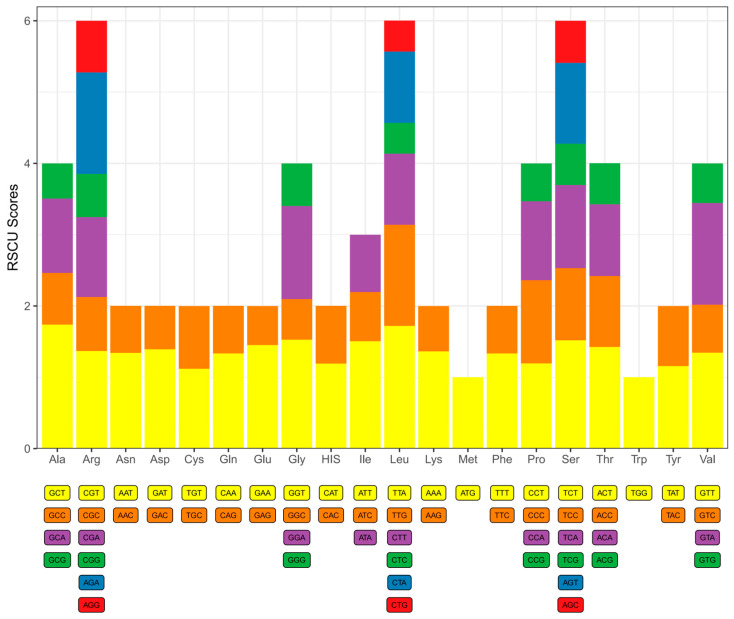
Relative synonymous codon usage (RSCU) in the plastome protein-coding genes of *Davallia trichomanoides*. Bars are grouped by amino acid. Within each group, stacked segments represent the RSCU value of each synonymous codon (labels shown below). Segment heights > 1.0 indicate over-represented codons and <1.0 indicate under-represented codons. Single-codon families (Met/AUG and Trp/UGG) show RSCU = 1.0.

**Figure 4 genes-16-01310-f004:**
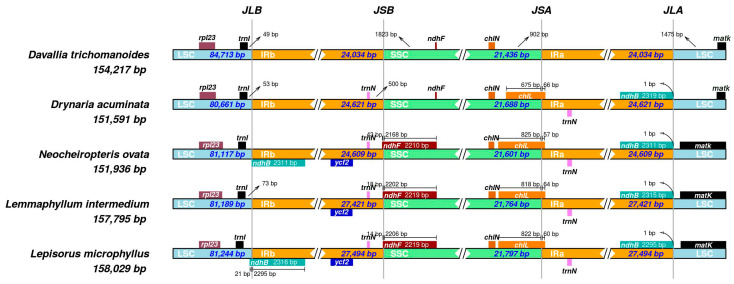
Comparison of the four junction sites (JLB, JSB, JSA, and JLA) between the inverted repeats (IRa, IRb) and single-copy regions (LSC and SSC) in five Polypodiaceae plastomes. Genes near the junctions are shown. Numbers indicate the distance in base pairs from the junction to the start or end of the adjacent gene.

**Figure 5 genes-16-01310-f005:**
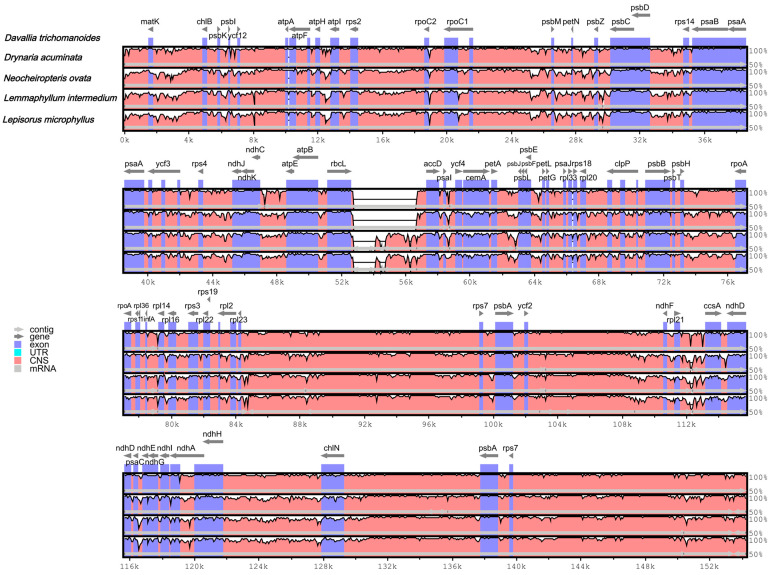
Whole-plastome alignment (mVISTA; reference *D. trichomanoides*). Gray arrows denote gene orientation, and thick black bars indicate the positions of the inverted repeats (IRs). The *y*-axis represents percent identity ranging from 50% to 100%. The color coding of the genomic region is protein coding (exon, purple), ribosomal RNA (rRNA, cyan), and conserved non-coding sequences (CNS, pink).

**Figure 6 genes-16-01310-f006:**
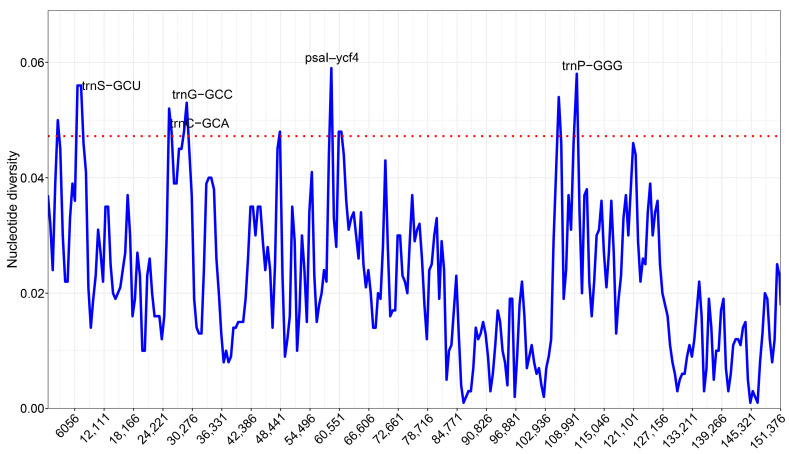
Sliding-window nucleotide diversity (Pi values) across five plastomes (window 600 bp; step 200 bp). The dashed line marks the 95th percentile. Labeled peaks denote hypervariable windows.

**Figure 7 genes-16-01310-f007:**
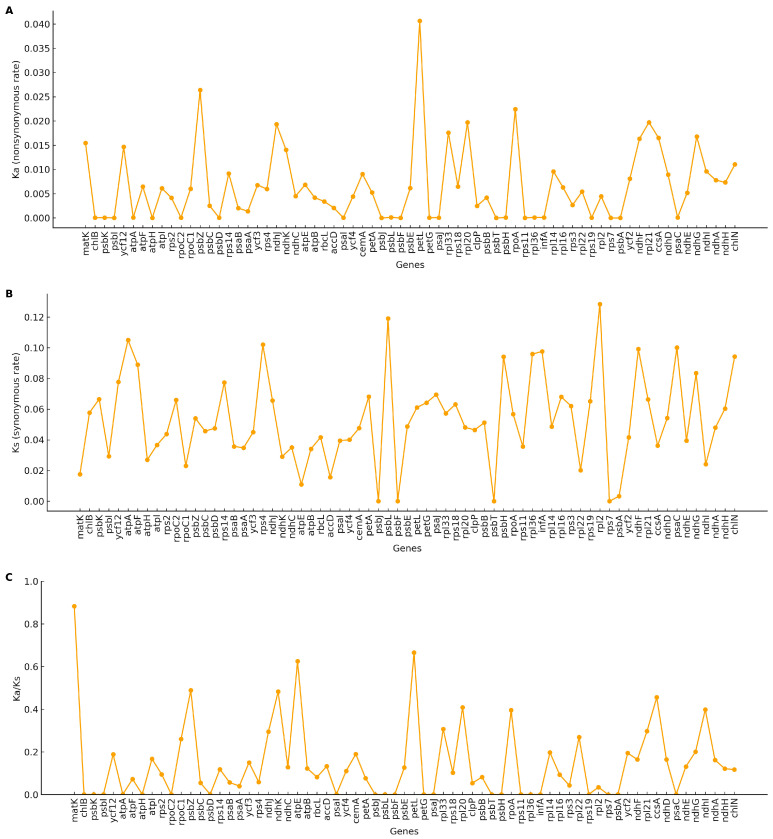
Selective pressure on protein-coding genes across five plastomes. (**A**) Nonsynonymous substitution rate (Ka) for each gene. (**B**) Synonymous substitution rate (Ks) for each gene. (**C**) Ratio of nonsynonymous to synonymous substitutions (Ka/Ks or ω). Each point represents a single gene. Genes with Ks = 0 were treated as NA; genes with saturated Ks (e.g., >2) were not interpreted.

**Figure 8 genes-16-01310-f008:**
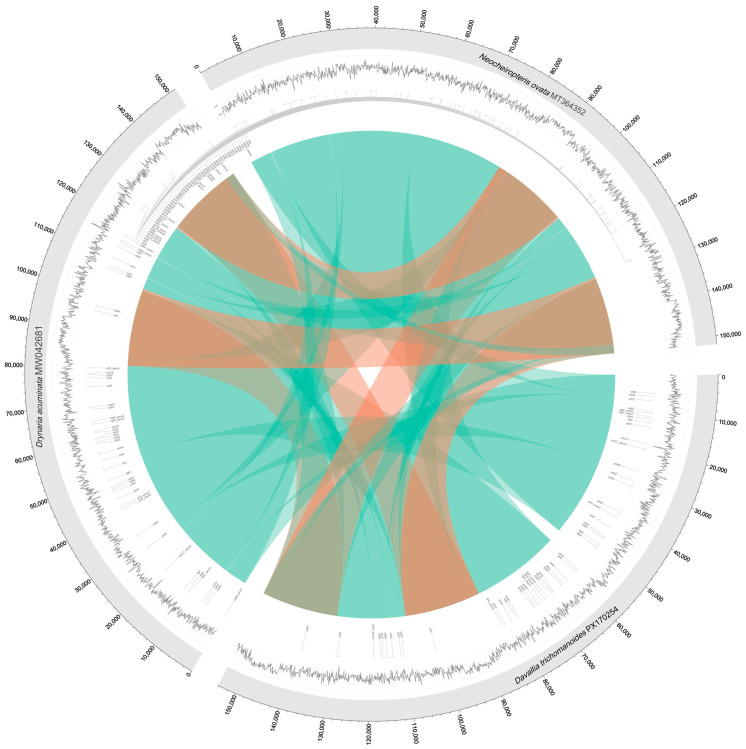
Circos plot visualizing synteny and alignment between three plastomes. The rings from outer to inner show: gene positions (outer ring), GC content (inner ring, black peaks), and aligned sequences between genomes (central ribbons). Ribbons are colored cyan for forward alignments and red for reverse alignments. Alignments were filtered for ≥80% identity and ≥500 bp length.

**Figure 9 genes-16-01310-f009:**
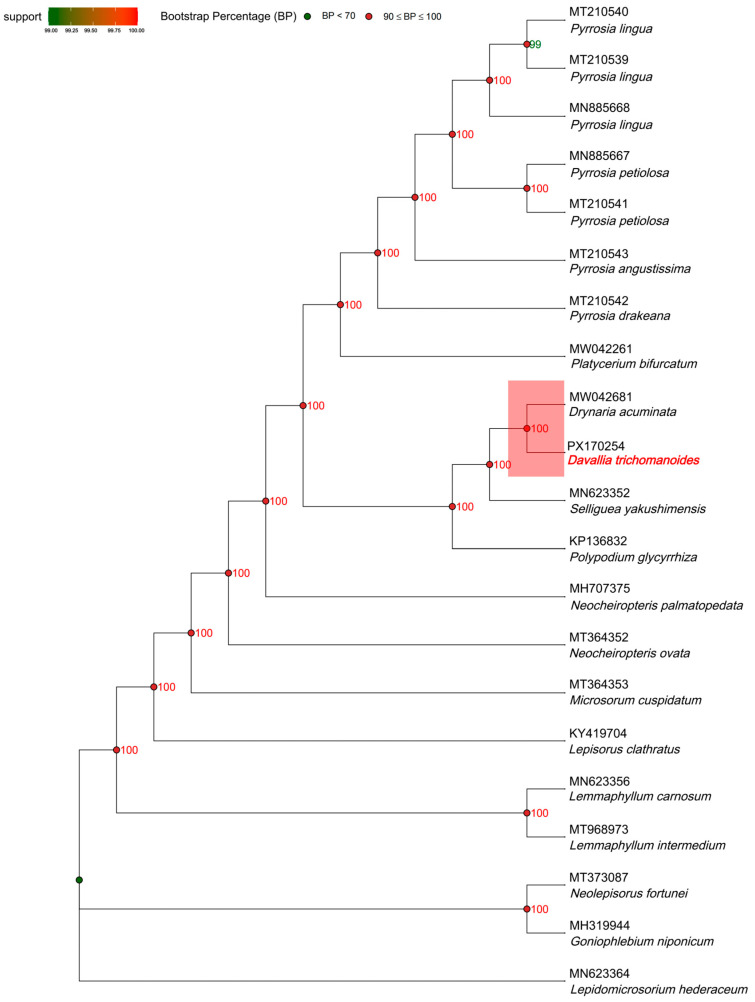
Maximum-likelihood plastome phylogeny of *Davallia trichomanoides* and related Polypodiineae taxa. Node labels indicate bootstrap supports (%) (1000 replicates) under the best-fit partitioned model selected by ModelFinder. The tree is rooted with *Lepidomicrosorium hederaceum*; branch lengths are scaled to substitutions per site. For clarity, only nodes with support ≥ 70 are labeled; lower-support nodes are omitted.

**Table 1 genes-16-01310-t001:** Gene composition in the plastome of *Davallia trichomanoides*.

Category	Subcategory	Genes (Examples)	No. of Genes
Photosynthesis	Photosystem I	*psaA*, *psaB*, *psaC*, *psaI*, *psaJ*	5
	Photosystem II	*psbA*, *psbB*, *psbC*, *psbD*, *psbE*, *psbF*, *psbH*, *psbI*, *psbJ*, *psbK*, *psbL*, *psbM*, *psbT*, *psbZ*	15
	NADH dehydrogenase	*ndhA* *, *ndhC*, *ndhD*, *ndhE*, *ndhF*, *ndhG*, *ndhH*, *ndhI*, *ndhJ*, *ndhK*	10
	Cytochrome b6f	*petA*, *petG*, *petL*, *petN*	4
	ATP synthase	*atpA*, *atpB*, *atpE*, *atpF* *, *atpH*, *atpI*	6
	Rubisco	*rbcL*	1
	Protochlorophyllide reductase	*chlB*, *chlN*	2
Genetic apparatus	Ribosomal proteins (LSU)	*rpl14*, *rpl16*, *rpl2* *, *rpl20*, *rpl21*, *rpl22*, *rpl23*, *rpl33*, *rpl36*	9
	Ribosomal proteins (SSU)	*rps11*, *rps12* **, *rps14*, *rps18*, *rps19*, *rps2*, *rps3*, *rps4*, *rps7*	10
	RNA polymerase	*rpoA*, *rpoC1* *, *rpoC2*	3
	rRNAs	*rrn16*, *rrn23*, *rrn4.5*, *rrn5*	8
	tRNAs	*trnA-UGC* *, *trnC-GCA*, *trnD-GUC*, *trnE-UUC*, *trnF-GAA*, *trnG-GCC*, *trnG-UCC* *, *trnH-GUG*, *trnI-CAU*, *trnI-GAU* *, *trnL-UAA* *, *trnL-UAG*, *trnM-CAU*, *trnP-GGG*, *trnP-UGG*, *trnQ-UUG*, *trnR-ACG*, *trnR-UCU*, *trnS-GCU*, *trnS-GGA*, *trnS-UGA*, *trnT-GGU*, *trnT-UGU* *, *trnV-UAC* *, *trnW-CCA*, *trnY-GUA*	32
Other/housekeeping	Misc.	*accD*, *cemA*, *clpP* **, *ccsA*, *matK*, *infA*	6
	Conserved hypothetical chloroplast ORF	*ycf12*, *ycf2*, *ycf3* **, *ycf4*	4
Total			115

* Gene with one intron; ** gene with two introns.

**Table 2 genes-16-01310-t002:** Intron-bearing genes in the plastome of *Davallia trichomanoides*, with exon and intron lengths.

Gene	Location	Exon1	Exon2	Exon3	Intron1	Intron2
*trnG-UCC*	LSC	31	42		894	
*atpF*	LSC	144	411		1275	
*rpoC1*	LSC	177	831		1743	
*trnL-UAA*	LSC	34	51		580	
*trnV-UAC*	LSC	25	38		705	
*rps12*	IRa	232	26	114	844	71,702
*clpP*	LSC	71	290	242	1064	1084
*rpl2*	LSC	284	64		1039	
*trnI-GAU*	IRb	35	37		1080	
*trnA-UGC*	IRb	37	36		877	
*ndhA*	SSC	558	558		2044	
*trnA-UGC-2*	IRa	37	36		806	
*trnI-GAU-2*	IRa	35	37		1010	
*trnT-UGU*	IRb	34	38		539	
*trnT-UGU-2*	IRa	34	38		609	
*ycf3*	LSC	124	188	162	1095	985

## Data Availability

The data presented in this study are openly available in GenBank of NCBI at https://www.ncbi.nlm.nih.gov (accessed on 28 October 2025) under the accession number PX170254. The associated BioProject, SRA, and Biosample numbers are PRJNA1303833, SRR34948204, and SAMN50537047, respectively.

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
