# Peer review of "Complete Chloroplast Genome and Phylogenomic Analysis of Davallia trichomanoides (Polypodiaceae)"

_genes, 2025, doi:10.3390/genes16111310_

Round 1

Reviewer 1 Report

Comments and Suggestions for Authors

In this manuscript, the authors sequenced the chloroplast genome of the fern species Davallia trichomanoides and performed comparative analyses with closely related fern species. I find that the manuscript is well written and easy to follow. I only have a few minor questions:

(1) Line 95, "Total genomic DNA was extracted from leaf tissue..." But chloroplast genome is generally not considered genomic DNA. Some clarification is needed.

(2) Figure 8, "Circos plot based on complete chloroplast genome alignment between five species." But I only see three species on the plot. Also, the font size in this figure is quite small and hard to read. Could you make them bigger?

Author Response

Comment 1: Line 95, "Total genomic DNA was extracted from leaf tissue..." But chloroplast genome is generally not considered genomic DNA. Some clarification is needed.

Response 1: Thank you for your valuable comment. We understand the concern regarding the use of the term "genomic DNA," as chloroplast genomes are typically distinct from nuclear genomic DNA. In this study, we specifically extracted "total DNA" from leaf tissue, which contains both nuclear and plastid (chloroplast) DNA. For the purpose of assembling the chloroplast genome, we focused on the plastid fraction of this total DNA. We have revised the text for clarity and to better reflect the nature of the DNA used in our study.

Comment 2: Figure 8, "Circos plot based on complete chloroplast genome alignment between five species." But I only see three species on the plot. Also, the font size in this figure is quite small and hard to read. Could you make them bigger?

Response 2: Thank you for pointing this out. We apologize for the oversight regarding the number of species in the Circos plot. Upon reviewing the figure, we see that only three species are included in the plot instead of the five as stated. We will revise the figure legend to reflect the correct number of species. Additionally, we will increase the font size in the figure for better readability, as per your suggestion.

Reviewer 2 Report

Comments and Suggestions for Authors

The authors sequenced and analyzed the complete chloroplast genome of Davallia trichomanoides. They compared it with four other fern species to study structural differences, gene arrangement, repeats, and evolutionary patterns. The study shows that the genome is mostly conserved, with small changes in IR boundary regions. The results support the close relationship between D. trichomanoides and D. acuminata and suggest useful DNA regions for further genetic studies.

Major comments 

- This study adds another complete chloroplast genome to fern research. However, similar work has already been done on related ferns. The authors should explain more clearly what is new about Davallia trichomanoides and what makes it different from earlier studies.

- More information about the sequencing coverage, assembly quality, and genome verification is needed. 

- The reasons for choosing specific comparison species should be explained clearly. 

- The authors should describe how they handled errors or uncertainty in Ka/Ks analysis.

- The figures are informative but lack details on statistical values like variability or error. -- Adding simple summary statistics and improving figure readability would help readers understand the results better.

- The discussion mostly repeats known findings about chloroplast structure. The authors could relate their results to the ecology of epiphytic ferns. For example, they might discuss how structural stability relates to light or moisture conditions in their environment. 

- The phylogenetic tree fits known fern relationships. Adding nuclear data would make the phylogeny stronger. The authors could also mention the limitations of using only chloroplast genomes for evolutionary inference.

Minor comments 

- Sentences should be shorter and simpler. 

- Use active voice in the methods section to improve clarity. 

- Increase label size for clarity. 

- Simplify Table 1 by grouping genes into broad categories like "Photosynthesis" or "Ribosomal". 

- Check formatting and ensure all software tools have citation details. 

- Update older references if recent studies are available on fern plastome evolution.

- The authors should provide detailed sequences, read statistics, and alignment files as supplementary materials.

- The title is accurate but could be more specific by including “phylogenomic analysis.” -Adding more detailed keywords could help readers find the paper easily.

The manuscript presents valuable data and analysis but needs clarification, simplified language, and more focused discussion. 

Author Response

Comment 1. This study adds another complete chloroplast genome to fern research. However, similar work has already been done on related ferns. The authors should explain more clearly what is new about Davallia trichomanoides and what makes it different from earlier studies.

Response 1: Thank you for the suggestion. We have revised the Introduction to state the novelty explicitly: to our knowledge this is the first complete plastome of Davallia trichomanoides, whereas allied epiphytic Polypodiineae already have published plastomes (e.g., Drynaria acuminata, Lemmaphyllum intermedium, Neocheiropteris palmatopedata, and Leptochilus/Lepisorus spp.). Our study closes that taxon gap and tests whether IR size class and SSC/IR boundary features reported in those genera also characterize Davallia, while providing new comparative resources (repeats/SSRs, π hotspots, Ka/Ks, and synteny).

Comment 2. More information about the sequencing coverage, assembly quality, and genome verification is needed.

Response 2: Thank you for this helpful suggestion. We now report data-quality and validation metrics: Illumina PE sequencing yielded 93,481,172 reads (13.93 Gb) with high base accuracy (Q20 = 98.84%, Q30 = 95.94%) (Table S1). The 154,217-bp plastome was assembled and circularized with GetOrganelle. Mapping all clean reads back to the assembly showed a mean depth of 535.65× with 100% bases ≥1× (154,217/154,217), ~100% ≥10× (154,202), 99.9% ≥50× (154,106), 99.1% ≥100× (152,802), and 92.4% ≥200×, with no coverage gaps (Table S2; Figure S1). Read-mapping confirmation is standard practice for plastome validation, and GetOrganelle is widely used for accurate plastome assemblies; together these results support the completeness and accuracy of our chloroplast genome.

Comment 3. The reasons for choosing specific comparison species should be explained clearly.

Response 3: Thank you for this helpful suggestion. We selected Drynaria acuminata, Neocheiropteris ovata, Lemmaphyllum intermedium, and a representative of Lepisorus because they are epiphytic Polypodiineae that encompass the closest polypodiaceous relatives pertinent to Davallia's classification, and they display the IR-boundary dynamics that are characteristic and informative within this group. Recent plastid phylogenomics has clarified these relationships and highlighted lineage-specific IR variation, while complete, publicly available plastomes for these taxa enable comparable, reproducible comparisons.

Comment 4. The authors should describe how they handled errors or uncertainty in Ka/Ks analysis.

Response 4: Thank you for this helpful comment. We now clarify that Ka/Ks estimates were based on codon-aware alignments (protein-guided back-translation), with low-confidence columns masked (Gblocks) and problematic ORFs removed; Ka and Ks were computed with KaKs_Calculator 3.0, while Ks = 0 or saturated Ks genes were not interpreted, and ω > 1 was not considered evidence of positive selection without site-model support. These steps reduce alignment/model bias and provide conservative, reproducible ω estimates.

Comment 5. The figures are informative but lack details on statistical values like variability or error. -- Adding simple summary statistics and improving figure readability would help readers understand the results better.

Response 5: Thank you for this helpful suggestion. We have revised the figures to report simple variability/error metrics and to improve legibility. All plots were redrawn at high resolution with larger fonts and consistent legends to aid readability.

Comment 6. The discussion mostly repeats known findings about chloroplast structure. The authors could relate their results to the ecology of epiphytic ferns. For example, they might discuss how structural stability relates to light or moisture conditions in their environment.

Response 6: Thank you for this suggestion. We added a short paragraph (section 4.8) in the Discussion that links our plastome results to epiphytic canopy ecology and explains why structural stability is plausible in this niche.

Comment 7. The phylogenetic tree fits known fern relationships. Adding nuclear data would make the phylogeny stronger. The authors could also mention the limitations of using only chloroplast genomes for evolutionary inference.

Response 7: Thank you for your constructive suggestion. We agree that adding nuclear markers would improve the phylogeny and enable explicit tests of plastid–nuclear concordance. In this study, we focused on complete plastomes to clarify relationships within Polypodiineae and present the first plastome for D. trichomanoides. Generating comparable nuclear datasets across all taxa is beyond our current scope. We've included a brief statement on the limitations of plastid-only inference (uniparental inheritance, single non-recombining locus, potential introgression), and plan to integrate target-capture nuclear loci in future research to test concordance and enhance resolution. 

Comment 8. Sentences should be shorter and simpler.

Response 8: Thank you for the suggestion. We have revised the manuscript to use shorter, simpler sentences.

Comment 9. Use active voice in the methods section to improve clarity.

Response 9: Thank you for the suggestion. We have revised the Methods to active voice and shorter sentences for clarity.

Comment 10. Increase label size for clarity.

Response 10: Thank you for the helpful suggestion. We have redone figures at their final print size and enlarged the label and legend fonts for clarity. All figures are provided as ≥600 dpi TIFFs to ensure readability.

Comment 11. Simplify Table 1 by grouping genes into broad categories like "Photosynthesis" or "Ribosomal".

Response 11: Thank you for the helpful suggestion. We've simplified Table 1 by grouping genes into broad functional categories: Photosynthesis, Genetic apparatus, and Other/housekeeping.

Comment 12. Check formatting and ensure all software tools have citation details.

Response 12: Thank you for the reminder. We performed a manuscript-wide formatting audit and standardized tool names/versions and all citation details.

Comment 13. Update older references if recent studies are available on fern plastome evolution.

Response 13: Thank you for the suggestion. We updated the Introduction and Discussion with recent work on fern plastome evolution, replacing older citations.

Comment 14. The authors should provide detailed sequences, read statistics, and alignment files as supplementary materials.

Response 14: Thank you for the suggestion. We have submitted the requested materials as Supplementary files: the complete plastome sequence (FASTA) with annotated GenBank, read statistics and QC/mapping coverage (Table S1–S2; Figure S1), and the phylogenetic alignment matrices (FASTA). We also provide the GenBank accession and NCBI SRA information in the Data Availability statement, in line with MDPI Genes guidance on sharing full datasets and supplementary materials.

Comment 15. The title is accurate but could be more specific by including “phylogenomic analysis.” -Adding more detailed keywords could help readers find the paper easily.

Response 15: Thank you for the suggestion. We have revised the title to explicitly include “phylogenomic analysis” and expanded the keywords to improve searchability and indexing.

Round 2

Reviewer 2 Report

Comments and Suggestions for Authors

The study is well-organized and clear. It focuses on the chloroplast genome of Davallia trichomanoides and related ferns. The results and main points are easy to follow. The manuscript is suitable for publication in Genes.

Major comments

- The abstract is too long. It has 294 words. I recommend 200–250 words. Please shorten the abstract. Delete background/Objectives, Methods, Results in the abstract.

Minor comments

- Define all abbreviations when they first appear, even if they are in the abstract.

- Make sure tables are easy to read. Use clear borders and a simple font.

- Check that figure legends are easy to understand, especially for special plots.

- The English is good, but some sentences are very long. Use shorter sentences for better flow.

- Fix minor typos, such as missing spaces between numbers and units.

- After these changes, the manuscript will be ready for acceptance.

Author Response

Comment 1. The abstract is too long. It has 294 words. I recommend 200–250 words. Please shorten the abstract. Delete background/Objectives, Methods, Results in the abstract.

Response 1: We have shortened the abstract to approximately 170 words. Additionally, we have removed the words "Background," "Objectives," "Methods," and "Results" as per the reviewer’s request.

Comment 2. Define all abbreviations when they first appear, even if they are in the abstract.

Response 2: Thank you for your suggestion. We have revised the manuscript to define all abbreviations when they first appear, including in the abstract, as requested.

Comment 3. Make sure tables are easy to read. Use clear borders and a simple font.

Response 3: Thank you for your suggestion. We have made the necessary adjustments to ensure that tables are easier to read. Clear borders have been added, and a simple, legible font has been used throughout the tables to improve clarity.

Comment 4. Check that figure legends are easy to understand, especially for special plots.

Response 4: We thank the reviewer for this important suggestion to improve the accessibility of our figures. We have thoroughly revised the figure legends to enhance their clarity, specifically for the specialized plots, such as Figures 2, 4, 7, and 8.

Comment 5. The English is good, but some sentences are very long. Use shorter sentences for better flow.

Response 5: We thank the reviewer for this valuable suggestion to improve the readability of our manuscript. We have thoroughly reviewed the text and identified numerous lengthy or complex sentences. As recommended, we have revised these by breaking them into shorter, more direct sentences, particularly in the Introduction, Methods, and Discussion sections.

Comment 6. Fix minor typos, such as missing spaces between numbers and units.

Response 6: We thank the reviewer for this careful observation. We have thoroughly checked the entire manuscript and corrected all instances of missing spaces between numbers and their units.